# The Role of Artificial Intelligence and Machine Learning in the Prediction of Right Heart Failure after Left Ventricular Assist Device Implantation: A Comprehensive Review

**DOI:** 10.3390/diagnostics14040380

**Published:** 2024-02-09

**Authors:** Ozlem Balcioglu, Cemre Ozgocmen, Dilber Uzun Ozsahin, Tahir Yagdi

**Affiliations:** 1Department of Cardiovascular Surgery, Faculty of Medicine, Near East University, TRNC Mersin 10, Nicosia 99138, Turkey; ozlem.balcioglu@neu.edu.tr; 2Operational Research Center in Healthcare, Near East University, TRNC Mersin 10, Nicosia 99138, Turkey; dozsahin@sharjah.ac.ae; 3Department of Biomedical Engineering, Faculty of Engineering, Near East University, TRNC Mersin 10, Nicosia 99138, Turkey; cemreozgocmen@hotmail.com; 4Medical Diagnostic Imaging Department, College of Health Sciences, University of Sharjah, Sharjah 27272, United Arab Emirates; 5Department of Cardiovascular Surgery, Faculty of Medicine, Ege University, Izmir 35100, Turkey

**Keywords:** left ventricular assist device, right heart failure, right ventricle failure, artificial intelligence, machine learning

## Abstract

One of the most challenging and prevalent side effects of LVAD implantation is that of right heart failure (RHF) that may develop afterwards. The purpose of this study is to review and highlight recent advances in the uses of AI in evaluating RHF after LVAD implantation. The available literature was scanned using certain key words (artificial intelligence, machine learning, left ventricular assist device, prediction of right heart failure after LVAD) was scanned within Pubmed, Web of Science, and Google Scholar databases. Conventional risk scoring systems were also summarized, with their pros and cons being included in the results section of this study in order to provide a useful contrast with AI-based models. There are certain interesting and innovative ML approaches towards RHF prediction among the studies reviewed as well as more straightforward approaches that identified certain important predictive clinical parameters. Despite their accomplishments, the resulting AUC scores were far from ideal for these methods to be considered fully sufficient. The reasons for this include the low number of studies, standardized data availability, and lack of prospective studies. Another topic briefly discussed in this study is that relating to the ethical and legal considerations of using AI-based systems in healthcare. In the end, we believe that it would be beneficial for clinicians to not ignore these developments despite the current research indicating more time is needed for AI-based prediction models to achieve a better performance.

## 1. Introduction

Despite all the major pharmacological and clinical improvements in cardiovascular disease management, heart failure still remains a global public health concern, with dramatic increases annually [1]. Since ventricular assist device implantation began to be used for end stage heart failure patients, treatment strategies for heart failure have changed dramatically. Currently, left ventricular assist device (LVAD) implantation is the most accepted alternative treatment option of heart transplantation for the aforementioned patient group. As a result of the advancements in LVAD technology and experience in using it, 1-year survival after implantation has now increased to over 80% [2,3,4,5,6]. However, well-known and serious side effects still exist. One of the most challenging and prevalent side effects of LVAD implantation, which is the topic of our review, is that of right heart failure (RHF) that may develop afterwards and may cause significant early- and/or long-term detrimental effects.

RHF occurs in 10–40% of all cases after LVAD implantation [7,8,9,10,11,12,13,14,15,16,17,18,19,20,21,22,23,24,25]. The main reasons for this wide prevalence range that is reported in different studies are differences in the diagnostic criteria of RHF, patient demographic characteristics, and institutional management strategies [26,27]. Even though well-accepted signs and symptoms of RHF are known by all clinicians, there remain no universally accepted classification and definition algorithms. Certain studies are based on clinical statuses like high central venous pressure (CVP), the requirement of inotropic support, an increase in pulmonary arterial pressure, nitric oxide (NO) inhalation, or the requirement of mechanical support for the right ventricle [28,29,30,31,32,33,34].

Over time, certain changes have been made to eliminate ambiguity and inconsistency in the definition of RHF [35,36,37,38]. Recently, the Mechanical Circulatory Support Academic Research Consortium has proposed a broader, more comprehensive definition of RHF after LVAD implantation [39].

The fact that most studies are conducted in a single center and on relatively small patient groups reduces the reliability of the analyses of the pre- and postoperative results. The variable characteristics of LVADs and patient demographics also make meaningful interpretations difficult. Additionally, due to the complex and heterogeneous pathophysiology of RHF, it is not easy to classify or generate a risk score for post-LVAD RHF. However, risk scores created using multivariate analyses have provided significant benefits in determining the development of RHF. Thus, it would not be wrong to say that one of the most useful things to carry out in order to prevent RHF after LVAD implantation is to use a validated risk scoring system.

On the other hand, with the increasing use of AI and ML applications in the medical field in recent years, the doors of new development have been opened in this regard. The number of studies conducted on this subject is increasing year by year.

Being first proposed by Alan Turing in 1950, AI is the concept of creating a digital mind that can learn and “think” like a human mind can. ML is a process that is encompassed by AI, where a software model can learn to acquire new data and interpret them in a meaningful way for the task at hand, therefore providing useful feedback and surpassing classical algorithm-based computer programming. It is viable that AI could fulfill any and all given assignments as long as the ML model is sophisticated enough [40,41,42].

There are two main approaches to ML with many more subtypes and variations below them. “Supervised ML” involves identifying and labeling data with known class information and training the ML model with this available knowledge. Therefore, a ML model can then determine the common properties within each class and thus “learn” to identify that class on its own. Supervised ML, however, requires manual dataset preparation for labelling and needs expert knowledge about the task [43,44].

The second approach of “unsupervised ML” is carried out by making all the data available to the software model, withholding the information about classes, and letting the ML model decide which cases belong together according to their data similarities. Unsupervised ML can be very beneficial for obtaining new information about complex systems with many variables as clustering decisions and feature associations are made by the model on its own, sometimes yielding previously unexpected results. [43,44]. A flowchart visually explaining the ML process is shown in Figure 1.

Regarding our subject matter, it can be easily imagined that AI will be able to help in a multitude of ways, such as by discovering RHF mechanism–hidden parameter relationships, the early prediction of risk groups pre-implantation, the identification of clinical parameters that are critical for early RHF detection post-implantation, and determining treatment plans for RHF patients post-implantation in order to change their prognosis. Achieving all of these tasks through conventional means seems to be a difficult undertaking when the sheer volume of data analysis that would be necessary is considered. Therefore, utilizing AI for this job appears to be a natural fit [45,46,47,48].

The purpose of this study is to review and highlight recent advances in the uses of AI in evaluating RHF after LVAD implantation as these types of research will lead to a better understanding of a common issue with LVAD implantation, which is of critical importance.

## 2. Method

The available literature has been scanned using the following phrases: “artificial intelligence and left ventricular assist device”, “machine learning and left ventricular assist device”, “right heart failure and machine learning”, “prediction of right heart failure after LVAD”, “deep learning and LVAD”. Pubmed, Web of Science, and Google Scholar databases were used in order to search for published articles. In order to achieve a better understanding of the role of AI and ML techniques, we also reviewed studies relating to previous risk scoring systems and summarized them all with their pros and cons. All the published articles were analyzed by the authors of this review. We found 9 studies using conventional risk score systems (Table 1) and 9 studies using AI systems (Table 2) for the prediction of RHF after LVAD implantation. We believe this will provide a useful contrast with AI-based models and allow readers to compare these different approaches and their results to the same problem of RHF post LVAD implantation.

## 3. Results

### 3.1. Conventional Risk Prediction Scores

In order to more accurately predict the risk of developing RHF after LVAD implantation, risk scoring studies began to emerge after the 2000s. Successive studies conducted after 2008 have aimed to guide clinical practices in this regard. A significant part of these studies are single-center studies, and various clinical, hemodynamic, biochemical, and echocardiographic data were evaluated together to attain the most ideal scoring system (Table 1). The definition of RHF varies for each study. Additionally, there is no full consensus on the variables used to create the risk score. In 2008, Matthews et al. used multivariate logistic regression for patients who had mostly underwent pulsatile LVAD implantation [49]. The Michigan right ventricular failure risk score (RVFRS), which was developed by the authors, is the first model for the preoperative risk stratification of RV failure in LVAD candidates. An elevated ALT, vasopressor requirement, as well as high bilirubin and creatinine levels were predictors of RHF in multivariate analyses. These variables were used to create the risk scores. This model has been found to be very effective in predicting RHF after LVAD implantation. In the same year, the Penn RVAD risk score was created by Fitzpatrick et al. [50]. This study established a risk score by showing that a low preoperative cardiac index (CI) and right ventricular stroke work index (RVSWI), severe pre-VAD RV dysfunction, a high creatinine level, previous cardiac surgery, and hypotension all increase the risk of RHF after LVAD implantation. After analysis of this risk scoring, it was revealed that successful LVAD support was predicted for patients with a low score, while the probability of biventricular assist device (BiVAD) placement was high for patients with a high score.

One previous study, which used multivariate logistic regression analysis, pointed to three preoperative factors that seemed to be significantly associated with RVF after LVAD implantation as follows: (1) the need for intra-aortic balloon counterpulsation before the operation, (2) an increase in pulmonary vascular resistance, and (3) device implantation as a destination therapy [51]. The risk score (Utah RV risk score) was calculated as the sum of the points assigned for the existence of a certain perioperative variable (Table 1).

The developed RVF risk score effectively predicted the risk of RV failure. Additionally, the results revealed a significant reduction in survival at days 30, 180, and 365 after LVAD implantation using the risk score model.

In 2011, Kormos et al. evaluated the incidence, risk factors, and effect of the outcomes of right ventricular failure for patients who had been implanted with a continuous-flow LVAD (HeartMate II) [52]. Multivariate analysis showed that a central venous pressure/pulmonary capillary wedge pressure ratio (CVP/PCWP) higher than 0.63, the need for preoperative ventilator support, and a BUN level higher than 39 mg/dL were independent predictors of right ventricular failure after HeartMate II implantation. The authors also concluded that the rates of RVF and RVAD need that were observed for patients with the HeartMate II are low relative to the previous results with pulsatile LVADs and support the use of new-generation continuous-flow devices for end-stage heart failure.

In another study, Atluri et al. defined severe right ventricular dysfunction based on echocardiographic parameters, taking into account right ventricular contractility, tricuspid regurgitation, and tricuspid annular motion [53]. In multivariate logistic regression analysis, a CVP of > 15 mmHg, severe RV dysfunction, preoperative intubation, severe tricuspid regurgitation, and tachycardia were determined to be major criteria predicting the need for biventricular support. Based on this analysis, they established the CRITT score as a predictor of RVF. The ability to quickly calculate the CRITT score at the bedside without the need for complex calculations is an advantage that increases its applicability.

Many of the RV risk scoring systems that were first proposed did not use detailed imaging parameters to aid risk stratification. Raina et al. combined echocardiographic variables, such as right ventricular fractional area change (RV FAC), the left atrial (LA) volume index, and the estimated right atrial (RA) pressure with an echocardiographic scoring system to estimate RVF [10]. They concluded that combining the echocardiographic variables with a simple, easily interpreted echocardiographic scoring system significantly improved the prediction of RVF versus any one echocardiographic variable used to carry this out alone.

Afterwards, a study from Germany proposed the ARVADE score, which consists of echocardiographic parameters [54]. In this study, multivariable analysis identified an INTERMACS level 1, an Em/SLAT ratio of ≥18.5 (Em: pulsed Doppler transmitral E wave; SLAT: tissue Doppler lateral systolic velocity), and the basal a RVEDD of ≥50 mm (right ventricular end-diastolic diameter) as independent predictors. Authors concluded that the ARVADE score, when calculated as the sum of scores for one clinical and three echocardiographic measures reflecting LV global systolic and diastolic dysfunction and RV congestion, may estimate suitability for LVAD implantation.

In 2018, Loforte et al. introduced a simple and easily memorized risk stratification tool (ALMA score) to determine whether an isolated LVAD (continuous-flow device) implantation could be tolerated [55]. A five-point risk score was developed based on the clinical variables identified in the multivariate logistic regression analysis as follows: the destination therapy (DT) intention, a pulmonary artery pulsatility index (PAPi) of <2, a right ventricular stroke work index (RVSWi) of <300 mm of Hg/mL/m^2^, a RV/LV ratio of >0.75, and a model for end-stage liver disease excluding international normalized ratio (MELD-XI) score of >17. Based on this model, the authors recommended BiVAD for patients with a score of 4 or 5.

Historically, older RVF risk scores were developed in the era of pulsatile-flow LVADs. The lack of validation studies has made it difficult for these models to accurately predict RVF in the current continuous-flow LVAD population. To more accurately predict RVF, models that use retrospective, predominantly single-center, primarily continuous-flow LVAD data have been developed. However, a common shortcoming in both the old and new risk scoring models is that they are subject to limited external validation and have a modest predictive value.

In 2018, Soliman et al. developed and validated a simple score to predict early RHF after continuous-flow LVAD implantation in a large population from the EUROMACS database [56]. The EUROMACS-RHF risk score is composed of severe RV dysfunction, a ratio of RA/PCWP of ≥0.54, advanced INTERMACS classes of 1–3, a need for ≥3 intravenous inotropes, and hemoglobin of ≤10 g/dL. A composite 5-point score predicted early RHF after LVAD implantation; moreover, as the score increased, the risk of both RHF and mortality increased. They claimed that the EUROMACS-RHF risk score outperformed the previously published scores and the known individual echocardiographic and hemodynamic markers of RHF. Finally, they validated the risk model in the validation cohort. The c index was 0.70 in the derivation versus 0.67 in the validation cohort.

Early studies examining the risk factors associated with RVF and developing various risk scores were generally based on the weighted sum of 4–7 risk factors contributing modest sensitivity or specificity. In addition, accurate predictions for patients who are at risk of RVF after LVAD implantation depend on the multidimensional and variable interactions of many perioperative variables that cannot be adequately captured using traditional multivariate modeling techniques. As a result, generalized recommendations for patient selection that are obtained from relatively small single-center patient groups have limited usefulness in practice.

Prediction models that we have summarized so far were the conventional statistical analysis methods. As AI began proving itself more within healthcare, heart failure subgroup-specific research increased as well, where the considerable LVAD- and heart transplant subject-related AI literature began populating journals more and more. Although the studies that are mentioned above evaluated risk factors regarding post-LVAD RHF, the fact that this is a multifactorial problem makes it especially hard to effectively investigate this issue properly through conventional means. Due to this reason, AI and ML enable a more comprehensive avenue of research on this topic.

### 3.2. AI-Based Studies/Risk Scores

The use of Bayesian statistical modeling was proposed by Loghmanpour et al. to overcome the limited predictive capacity of risk scores obtained from existing multivariate analyses [57]. This recommendation of the authors is based on the hypothesis that it is essential to consider the relationships and conditional probabilities between independent variables to achieve satisfactory statistical accuracy. In this context, Bayesian network (BN) algorithms can account for the nonlinear interactions between variables by identifying groups of risk factors and their conditional interdependencies. The Bayesian models reported in this study are particularly suitable for combining large sets of risk factors because they are based on the conditional probabilities of the likelihood of RVF for a given combination of interrelated variables. The authors suggested that these algorithms better reflect the prioritization of dynamic clinical information when using data provided by the INTERMACS database. To the authors’ knowledge, this was the first report of a prognostic RVF model following continuous-flow LVADs using the INTERMACS database and adopting ML methods for statistical analysis. They extracted 34 preoperative variables from the INTERMACS database of 10,909 patients from 2006 to 2014 in order to predict RVF after LVAD implantation. The definition of RVF was based on the INTERMACS definition prior to 2014. Overall, 2024 patients were diagnosed with RVF (18.5%), 293 with acute (<48 h after implant) RVF (2.7%), 1036 with early (from 48 h to 14 days) RVF (9.5%), and *n* = 695 with late onset (>14 days) RVF (6.4%). Systolic PAP, pre-albumin, LDH, and RVEF parameters were found to have the most predictive value among all the preoperative variables. The authors acknowledged that a retrospective study with incomplete data was not ideal for a more detailed analysis where RVF severity could also be considered. Patients already who were considered too risky for LVAD implantation due to the possibility of RVF and who thus never received a LVAD were unavoidably omitted from the dataset, perhaps skewing results. The authors analyzed accuracy, the area under the ROC curve (AUC), sensitivity, and specificity, respectively. According to their findings, the AUC of the Bayesian model was 0.90 for acute RV failure, 0.84 for early RV failure, and 0.88 for late RV failure after LVAD implantation, which significantly outperformed all the previously published risk scores.

In a 2018 study, Samura et al. utilized a supervised ML model in order to predict right ventricular assist device (RVAD) requirements for patients that will undergo LVAD implantation [58]. They used 42 preoperative clinical and hemodynamic parameters of 115 patients that proceeded to be implanted with a continuous-flow LVAD between the years of 2013 and 2017. As a result of their study, five parameters were highlighted as having the highest predictive value as follows: the left ventricular end-diastolic dimension, the left ventricular end-systolic dimension, the left ventricular ejection fraction, the etiology of the dilated phase of hypertrophic cardiomyopathy, and the less-distensible right ventricle. Eight different ML algorithms were tested in order to obtain the best results, and they declared that a derived naïve Bayes model achieved a high level of accuracy of 95% and an area under the curve (AUC) value of 0.85. The researchers concluded that this method was useful and feasible in order to preoperatively predict which patients would likely need RVAD implantation.

Bellavia et al. used the ML approach to find out the association between the regional right ventricular and right atrial strains for the prediction of right ventricular failure in both the early and late postoperative period [59]. The Michigan risk score along with CVP and the apical longitudinal systolic strain of the right ventricular free wall were found to be the most important predictors of acute RHF. For chronic RHF, the most prominent predictors were those of right ventricular free wall systolic strain of the middle segment, right atrial strain, and tricuspid annular plane systolic excursion.

Shad et al. used a combination of greyscale video data and optical flow streams from the video data with a three-dimensional 152-layer deep learning ML algorithm. A total of 1909 scans from 723 patients were evaluated in order to predict RHF development in LVAD patients [60]. The researchers used two clinical risk score systems, which were those of the CRITT and PENN scores, to identify patients who were potentially at risk of RHF after LVAD implantation. Subsequently, they compared the deep learning and ML system performance to the risk scores. The study included 941 LVAD patients who were separated into two groups as follows: group one (*n*:182) with RHF and group two (*n*:541) without RHF. Researchers found that the calculated area under the curve (AUC) using the CRITT and PENN scores were 0.616 and 0.605, respectively. The AUC of their AI system was reported to be 0.729, which means that the newly developed deep learning system performed with a higher level of accuracy in predicting RHF. Although they worked with a small and limited dataset, the results were challenging; they believed that AI would find a wider working place in relation to cardiovascular disease on account of this, especially when carrying out prediction studies. They further argued that when RVAD implantation is planned beforehand, which may perhaps be planned concurrently with LVAD implantation as opposed to using emergent RVAD implantation after the patient’s condition deteriorates, being able to predict the eventual development of RHF in patients before LVAD implantation may improve patient survivability as a result.

In 2021, Kilic et al. utilized extreme gradient boosting, which is an ensemble ML algorithm, in order to investigate the preoperative data’s association with postoperative adverse events, which translates into 90-day and 1-year survival rates [61]. This study involved 16,120 patients from 170 centers, with the dataset being acquired from the INTERMACS database. It includes patient demographics, comorbidities, laboratory parameters, clinic visit measurements, interval events during hospitalization prior to LVAD insertion, and concomitant operative procedures. Post-LVAD data collected in the INTERMACS database include those relating to adverse events and survival. Examples of these adverse events include thrombosis, RHF, infection, and bleeding. Reportedly, the end result of this study found that there was an improvement of 48.8% (*p* < 0.001) in the 90-day mortality prediction and an improvement of 36.9% (*p* < 0.001) in the 1-year mortality prediction through ML compared with usual logistic regression data analysis. ML models derived using the XGBoost algorithm were well calibrated and had an improved level of discrimination over logistic regression. Based on these findings, they concluded that ML may have an important role in risk prediction in LVAD treatment both independently and in addition to traditional modeling approaches such as logistic regression. Further study that focuses on specific adverse event prediction, such as RHF, may be conducted in order to better understand the underlying mechanisms of these clinical outcomes, which would translate into creating a better patient treatment plan accordingly.

Using the statistical computing tool called “R”, Kilic et al. evaluated data from ENDURANCE trials in 2020, which included 564 patients [62]. This study aimed to analyze the risk of major adverse events after LVAD implantation and how they transitioned into each other. These events were device malfunction, bleeding, infection, neurological/renal/respiratory dysfunction, and RHF. They identified that the most common adverse events were bleeding and infection. Interestingly, they found that RHF is one of the top three adverse events that leads to further adverse events most often, with bleeding and infection being the other two. The highest transition probabilities were found to be those of infection to infection (0.34), bleeding to bleeding (0.31), RHF to bleeding (0.31), RHF to infection (0.28), and bleeding to infection (0.26). Additionally, they found that RHF had the lowest median time to the first adverse event, which was 3.5 days. Highlighting the importance of RHF for overall mortality rates post-LVAD implantation, patients with RHF were shown to have 50% mortality rates. RHF was also identified to be significantly linked with bleeding and infections, and it then follows that RHF prediction is vital for the successful survivability of long-term LVAD patients.

A total of 2550 patient data from the International Registry for Mechanically Assisted Circulatory Support (IMACS) database were utilized by Nayak et al. in 2022 in order to analyze 41 pre-implant variables of patients with acute post-LVAD RHF [63]. An unsupervised ML model was used in their work, identifying four RHF phenotypes, with the severe shock phenotype having the worst clinical outcomes. Ischemic cardiomyopathy (ICM) with a low-grade shock and non-ICM without any shocks were the two other phenotypes identified. The best clinical outcomes were observed for ICM without any shock phenotypes. The notion of classifying patients into phenotypes may prove useful for future research as applying separate ML-based prediction or analysis models to the significantly differing pathophysiologies of RHF could improve the predictive capabilities of pre-implant evaluations overall.

The study that was designed by Bahl et al. was one of the newest studies that focused on ML and RHF [64]. They preferred an “explainable” ML method called boosted decision trees in order to analyze the preimplant patient factors in nonlinear interactions with RHF after LVAD implantation. The study includes patients in the INTERMACS database who were implanted with their first durable LVAD between 2008 and 2017. A total of 186 potential risk factors were analyzed from 19,595 patients as unbiasedly and as comprehensively as possible. This study was aimed at better quantifying and understanding how different clinical variables both impact each other and the complex mechanism that leads to RHF after LVAD implantation. The study showed that in 19.1% of patients, severe RHF developed within the first 30 days. Thirty top predictors of RHF were identified. The INTERMACS profile, model for end-stage liver disease score, number of inotropic infusions, hemoglobin, and race were the first five top factors. Additionally, many of these top factors showed nonlinear relationships with key risk inflection points such as an INTERMACS profile of 2–3, right atrial pressure of 15 mmHg, pulmonary artery pressure index of 3, and prealbumin level of 23 mg/dL. They claimed that ML offers a number of algorithms that are far more flexible and are well equipped for high-dimensional, nonlinear, interacting relationships. They also believed that this study could open a new era for researchers to formulate patient optimization strategies before LVAD implantation.

Using a convolutional neural network (U-Net), Just et al. evaluated the preoperative CT scan data of 137 patients in order to assess their body composition and then predict major postoperative complications after LVAD implantation [65]. Body composition evaluation included the visceral and subcutaneous adipose tissue areas, psoas, total abdominal muscle areas, and sarcopenia. The body composition parameters were correlated with the major postoperative complication rates, such as postoperative infections, in-hospital mortality, and overall quality of life. They found that the adipose tissue distribution/concentration was an effective predictor of postoperative infections, in-hospital mortality, an impaired 6 min walking distance, and quality of life within 6 months postoperatively. While the study focused on all the causes related to the outcome prediction, RHF was one of the poor outcome classes present in the dataset. Therefore, a focus study on the usefulness of AI in RHF prediction using a similar dataset might be warranted. The method and performance summary of the reviewed publications are presented in Table 2.

## 4. Discussion

Due to the high morbidity and mortality of RHF after LVAD implantation, effective treatment methods are quite comprehensive and depend on center experiences and opportunities. The initial therapy after post-LVAD RHF starts is that using pharmaceutical treatment (inotropic support, NO inhalation, and forced diuresis). If RHF still remains after efficient medical treatment, mechanical device support comes into consideration even with the accompanying high mortality rates [66,67,68,69,70,71,72].

Therefore, it is critical to decide whether the patient needs BVAD before or during surgery at the latest. However, despite the preparations and precautions taken by evaluating many clinical, hemodynamic, biochemical, and echocardiographic criteria, it is not always possible to make the right choice between LVAD and BVAD. In order to make the right decision, it is important to thoroughly evaluate the right ventricular anatomy and functions before surgery [73,74,75,76,77,78,79,80,81,82,83,84,85,86,87,88,89,90,91,92,93,94,95,96,97,98,99,100].

Previously published prediction models were used to find out the link between post-LVAD RHF and the possible risk factors such as those in linear interactions.

However, just like RHF does, certain clinical situations usually present complex clinical aspects that may force the clinicians to think in a versatile manner to improve the outcomes. ML is a new, challenging method that has started to be used for cardiovascular diseases, especially for decision making, prediction scores, and prognostication. Compared with conventional statistical methods that work to find out the correlations between risk factors and the outcomes through mathematical equations, ML aims to discover the association between multiple variables and outcomes using observations for linear and nonlinear interactions. The advantages of the technique have covered most of the gaps in the research field and provide more opinions for future studies.

In this comprehensive review, we analyzed and summarized published studies that had already taken place in the literature and aimed to report the prediction of RHF after LVAD implantation using ML.

As per the details of the abovementioned studies that we have summarized, there are certain points that should be emphasized in order to clarify the aims of ML approaches. For instance, the design of the studies is the most important checkpoint that increases their value. The results of the multi-centered planned studies appear to be stronger for the researchers. For this purpose, Kiliç et al. conducted multi-centered research that included more than 15,000 patients’ data that had been acquired from the INTERMACS database [61]. It is not easy for statistical methods to compare very highly variable factors with each other in an unbiased way in order to find the ones with the most predictive value. Another favorable feature of AI studies is determined using echocardiographic scans. Shad et al. reported that the data obtained with the help of AI performed better compared with the manually obtained echocardiographic measurements and clinical risk scores (CRITT and Penn) [60]. An AI-based system directly analyses the spatiotemporal information from the cardiac walls and valves instead of carrying out a segmental evaluation of the cardiac chambers. Consequently, the misleading results that were obtained through manually calculating cardiac functions were eliminated. While they used the largest dataset of echocardiography, they also noted certain limitations. Because the study was planned retrospectively, the acquisition of the echocardiography dataset remains unstandardized. Additionally, the timing of the last echocardiographic scans was not carried out following the same timeline for all the patients. Although the key point of the study was the focus that was paid to the pre-training large video dataset, it was noted that prospective evaluations and timeline standardization would result in much better estimations.

In addition to the echocardiographic images that were used in making predictions for LVAD patients, the Berlin Heart Center researchers reported an interesting study designed to find out the relationship between body composition and postoperative LVAD complications using AI techniques. A total of 137 patients were included in the study who had underwent a CT scan before surgery. The AI-based evaluation of their body composition demonstrated that a higher number of patients with higher levels of visceral adipose tissue and subcutaneous adipose tissue suffer worse postoperative outcomes. This study provided the opinion of using CT scan images for carrying out prediction analysis of LVAD patients [65].

As we have analyzed in this review, ML models showed a promising level of discrimination in predicting RHF after LVAD implantation. The majority of the AI studies achieved AUC rates that were above 0.6 when compared to contemporary clinical risk scores. ML-based AI prediction models allow clinicians to make more efficient assessments and gain more predictive insight into whether a patient will suffer from RHF.

Despite all of this, there still needs to be a discussion about a few glaring issues that were common in most of the evaluated publications. For example, despite their accomplishments, the resulting AUC scores were far from ideal in order for these methods to be considered sufficient. We are aware that this is due to a number of results, one of which is that this type of research can still be considered to be in its infancy; the performance of these models will improve over time with each attempt and iteration.

Another reason for this performance issue could be that although a lot of the reviewed studies agree on one thing, neither the methods nor the time frames of the obtained data are standardized across institutions nor is the availability of data variation sufficient. As is stated ad nauseam across different studies on AI, data availability and standardization are vital for any future model to be successful. Therefore, further steps towards this end must be very seriously considered.

Perhaps one last statement could be made about the lack of prospective studies. Almost all the currently available studies were carried out retrospectively. Although they are very valuable in their own right, it might be the case that carrying out more prospective studies could lead to better standardized data collection methods and the eventual data richness that is sought after by many more scientists who are working in this field. These are outcomes that may be accomplished alongside showcasing the predictive powers achievable through AI.

However, by relying on all the reviewed studies, it is more than safe to say that ML techniques have the potential to improve clinical decision making and patient selection in order to achieve much better clinical outcomes.

## 5. Limitations

Our review has certain limitations. One of these limitations is that this review is not systematic and was intended to be a literature review of this topic. Our research only included the medical databases that are mentioned above. Therefore, we understand that there are many other databases, and it is probable that paramedical publications in other scientific fields like biomedical engineering and bioinformatics may exist.

## 6. Conclusions

RHF is a severe problem that can occur after LVAD implantation. Being able to make accurate predictions for potential LVAD patients will surely help to decrease the incidence rate of RHF and reach better clinical outcomes for these patients. Nowadays, classical statistical methods have started to give way to AI techniques, which are better suited for multivariable analysis instead of linear evaluation studies.

As we can conclude from this review, AI techniques have begun to find their place in many different healthcare fields as well as in more specific subjects of medicine. We believe that it would be beneficial for clinicians to not ignore these developments, despite the current research indicating that it more time is needed for AI-based prediction models to achieve a better performance and become truly game changing.

There are certain interesting and innovative approaches used for RHF prediction among the reviewed studies. Shad et al. used echocardiography video data in order to predict RHF and Just et al. used CT scans to detect adipose tissue within the body, which was reportedly a useful indicator of major complications [60,65]. Another interesting approach was the one undertaken by Bellavia et al. where they combined a more limited set of preoperative parameters through both cathlab measurements and echocardiography measurements [59]. Far from discounting more conventional approaches, the reviewed studies identified certain important predictive clinical parameters with positive results. For example, Loghmanpour et al. found that the systolic PAP, pre-albumin, LDH, and RV EF were most important predictive parameters, and they achieved an impressive AUC rate in their study [57]. Similarly, Kilic et al. and Bahl et al. identified more predictive parameters and non-linear relationships between preoperative clinical parameters, which is one of the main strengths of AI-based investigations as it is very difficult to infer these relationships through statistical analysis [61,62,64].

Looking at all of the different approaches of the mentioned studies, they seem to indicate that an ideal hypothetical AI system should be utilizing all sorts of data types in order to be able to converge on the correct answer and make the most accurate predictions.

Although they currently have certain obvious limitations at this point in time, we believe that ML procedures will also improve alongside advances in AI and computational science. We should not, however, ignore the fact that a striking portion of the current difficulties on this subject seemingly lies in the lack of prospective studies that would generate a standardized methodological set of conventional preoperative data as well as more specific data types for novel ideas.

It might be also true to say that although similar clinical measurements are taken across most institutions for LVAD recipient patients, each institution follows their own “best” time frames and techniques for obtaining these data, which makes it much more difficult for AI-based models to function with the highest accuracy possible. We imagine that a multi-centered agreement on a set of guidelines for data collection purposes would be highly beneficial for researchers and patients alike.

One last point of consideration must also be highlighted, which is that of the issue of the ethical and legal responsibility of using AI in healthcare. As AI becomes more and more capable, instead of asking if something is possible, we need to start considering how such a tool should be utilized. What should be undertaken if AI predictions and risk score predictions do not align? Which one should be followed? We know that it is very difficult to know how the system works with certain forms of ML methods. What steps should be taken to ensure transparency for the sake of being able to trust the prediction system? How can we know what to undertake if AI predictions eventually make a mistake? As is widely known, medical malpractice issues are taken very seriously by all the parties involved. Therefore, who should be held responsible? It may seem very early to ask these questions, but we believe that asking them now and coming up with solutions will prove to be much better than waiting for these issues to show up in real-world cases.

The current number of published studies seems to indicate that this topic is just starting to capture the interest of researchers, and, in our opinion, more studies will be designed using AI techniques in the near future, eventually providing doctors with a valuable tool to tackle a significant problem with LVAD implantation.

## Figures and Tables

**Figure 1 diagnostics-14-00380-f001:**
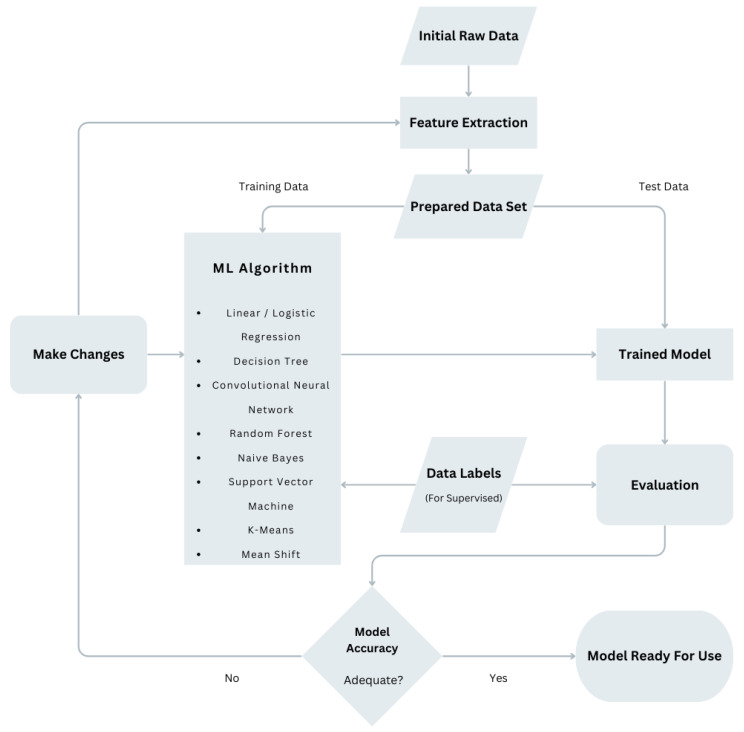
A flowchart visually explaining the machine learning process. Abbreviations: ML: machine learning.

**Table 1 diagnostics-14-00380-t001:** Comparison of studies using conventional risk prediction scores for RVF after LVAD implantation.

Study (Author, Year, RVF Risk Score)	Sample (*n*, Center, Device Type)	Study Design	Definition of RVF (Rate)	RVF Risk Score Component	Results
-Matthews et al. [49] -2008 -Michigan RV risk score	-197 patients -Single center -Mostly pulsatile	-Retrospective analysis of a prospectively collected LVAD database -MLR	35% -Need for postoperative IV inotrope support for >14 days -Inhaled NO for ≥48 h -Right-sided circulatory support -Hospital discharge on an inotrope	-Vasopressor requirement → 4 -AST being ≥80 IU/l → 2 -Bilirubin being ≥2.0 mg/dL → 2.5 -Creatinine being ≥2.3 mg/dL → 3 -Renal replacement therapy → 3	-RVF developed in 80% of patients with a RVFRS of ≥5.5 -AUC of a RVFRS was 0.73 ± 0.04
-Fitzpatrick et al. [50] -2008 -Penn RVAD risk score	-266 patients -Single center -Mostly pulsatile	-Retrospective -MLR	37% -Preoperative RV dysfunction grade -None → mild → moderate → severe -As determined from the echocardiogram performed prior to LVAD insertion.	18. (CI) + 18. (RVSWI) + 17. (creatinine) + 16. (previous cardiac surgery) + 16. (RV dysfunction) + 13. (systolic blood pressure)	-Score of ≥50 predicting the need for BiVAD -83% sensitivity and 80% specificity
-Drakos et al. [51] -2010 -Utah RV Risk Score	-175 patients -Single center -Mostly pulsatile	-Retrospective analysis of a prospectively collected LVAD database -MLR	44% -Need for inhaled NO for >48 h -IV inotropes for >14 days RVAD implantation	-DT → 3.5 -IABP → 4 -PVR of ≥4.3 WU → 4 -Inotropic dependency → 2.5 Beta-blocker → 2 Obesity → 2	-RVF% for risk score: ≤5.0 = 11% 5.5–8.0 = 37% 8.5–12.0 = 56% ≥12.5 = 83% -The AUC of the risk score was 0.743 ± 0.037
-Kormos et al. [52] -2011 -Kormos (HM II) RVF risk score	-484 patients -Multicentre -HM II	-Retrospective -MLR	20% -Requiring a RVAD -IV inotropes for >14 days after implantation -And/or inotropic support starting more than 14 days after implantation	-CVP/PCWP of >0.63 -Need for preoperative ventilator support -BUN level of >39 mg/dL	-Survival for patients without RVF at 180 days: 89% -Survival for patients with RVF at 180 days: 71% -AUC of the risk score was 0.68
-Atluri et al. [53] -2013 -CRITT score	-218 patients -Single center -HM II, pulsatile	-Retrospective -MLR	Criteria for initial BiVAD (rate 23%) Based upon ECHO parameters -RV contractility -Tricuspid regurgitation -Tricuspid annular motion	-CVP of >15 mmHg → 4 -Severe RVD → 2 -Intubation preoperatively → 2.5 -Severe TR → mg/dL → 3 -Tachycardia of >100 bpm → 3	-93% of patients with a score of 1 or less underwent successful isolated LVAD -80% of patients with a score of 4 or higher required BiVAD -AUC of the risk score was 0.80 ± 0.04
-Raina et al. [10] -2013 TTE Score	-55 patients -Single center -Mostly HMII	-Retrospective -MLR	29% -Need for inotropes for ≥14 days. -Need for temporary RVAD Criteria for initial BiVAD (rate of 23%) -Severe RVD on TTE -Severe PHT with a PVR of >5 WU or RAP of >15 mmHg -Sustained VA causing hemodynamic compromise	-LA volume index of <38 mL/m^2^ → 3 -RV FAC of <31% → 2 -RA pressure of >8 mm Hg → 2	-Score of ≥5 had a sensitivity of 63% and a specificity of 78% in predicting RVF
-Aissaoui et al. [54] -2015 -ARVADE score	-42 patients -Single center -Mostly HM II, HeartWare HVAD, pulsatile	-Prospective -MLR	57% -Need for placement of a temporary RVAD -Use of inotropic agents for 14 days	-Em/SLAT of ≥18.5 → 3 -RVEDD of ≥50 mm → 2 -INTERMACS level 1 → 1.5	-ARVADE score of >3.0 was predictive of post-LVAD RVF -Sensitivity of 89% and specificity of 74%
-Loforte et al. [55] -2018 -ALMA Score	-258 patients -2 centers -Mostly HM II, HeartWare HVAD, HM 3	-Retrospective -MLR	55% -Receiving short- or long-term RVAD despite maximal dosage of continuous inotropic support and NO ventilation	-DT → 1 -PAPi < 2 → 1 -RV/LV ratio of >0.75 → 1 -RVSWi of <300 mmHg/mL/m^2^ → 1 -MELD-XI score of >17 → 1	-Rate of RVF was 9% for a score of <2 -Rate of RVF was 57% for a score of 2–3 -Rate of RVF was 100% for a score of 4–5 -A score of 3 points provided sensitivity and specificity higher than 80%
-Soliman et al. [56] -2018 -EUROMACS-RHF risk score	-2988 Patients EUROMACS Database -Multicentre -Continuous-flow HeartWare HVAD, HM II, HM 3	-Retrospective analysis of the EUROMACS database -MLR	21.7% -Receiving short- or long-term RVAD support -Continuous inotropic support for ≥14 days -NO ventilation for ≥48 h	EUROMACS-RHF risk score/after adding CPB time -Need for ≥3 inotropic agents → 2.5/2 -INTERMACS class 1–3 → 2/2 -Severe RVD on the semiquantitative ECHO → 2/1. -RA/PCWP of >0.54 → 2/1 -Hemoglobin being ≤10 g/dL → 1/1.5 -CPB time being >100 min → -/1	-RHF risk ranged from 11% (low risk score of 0–2) to 43.1% (high risk score of >4 -AUC of the risk score was 0.75, 0.66, and 0.60 in the HM II, HeartWare HVAD, and HM 3

**Abbreviations:** RVF: right ventricular failure; LVAD: left ventricular assist device; MLR: multivariate logistic regression analysis; IV: intravenous; NO: nitric oxide; AST: aspartate aminotransferase; RVFRS: right ventricular failure risk score; AUC: area under the ROC curve; RVAD: right ventricular assist device; CI: cardiac index; RVSWI: right ventricular stroke work index; BiVAD: biventricular assist device; DT: destination therapy; IABP: intra-aortic balloon pump; PVR: pulmonary vascular resistance; HM II: HeartMate II; CVP: central venous pressure; PCWP: pulmonary capillary wedge pressure; BUN: blood urea nitrogen; ECHO: echocardiography; RVD: right ventricular dysfunction; TR: tricuspid regurgitation; TTE: transthoracic echocardiography; PHT: pulmonary hypertension; PVR: pulmonary vascular resistance; WU: wood unit; PAP: pulmonary artery pressure; VA: ventricular arrhythmia; LA: left atrium; RV FAC: right ventricular fractional area change; RA: right atrium; Em/SLAT: pulsed Doppler transmitral E wave/tissue Doppler lateral systolic velocity; RVEDD: right ventricular end-diastolic diameter; INTERMACS: Interagency Registry for Mechanically Assisted Circulatory Support; HM3: HeartMate 3; PAPi: pulmonary artery pulsatility index: MELD-XI: model for end-stage liver disease excluding the international normalized ratio; EUROMACS: European Registry for Patients with Mechanical Circulatory Support; CPB: cardiopulmonary bypass.

**Table 2 diagnostics-14-00380-t002:** Method and performance summary of the reviewed AI publications.

Authors (et al.)	Year	Title	Data Source	Findings
Loghmanpour [57]	2016	A Bayesian model to predict RVF following LVAD Therapy	INTERMACS data Patients: 10,909	Systolic PAP, pre-albumin, LDH, and RV EF are the most predictive preoperative variables. AUC of acute, early, and late RHF predictions is between 0.83 and 0.90 with a sensitivity of 90%
Samura [58]	2018	Prediction of RVF after left LVAD implantation using ML for preoperative hemodynamics	Preoperative clinical and hemodynamic parameters Patients: 115	Prediction accuracy is 95%, AUC is 0.85
Bellavia [59]	2020	Usefulness of regional RV and right atrial strain for the prediction of early and late RVF following a LVAD implant: a ML approach	Biomarkers, echocardiography, cath-lab measurements Patients: 74	Significant predictors: Michigan risk score, CVP, and systolic strain of RV free wall. ROC AUC is 0.95
Shad [60]	2021	Predicting post-operative RVF using video-based deep learning	Preoperative echocardiography video Patients: 723	ML AUC is 0.729, CRITT AUC is 0.616, Penn AUC is 0.605
Kilic [61]	2021	Using ML to improve risk prediction about durable LVAD implantation	INTERMACS data Patients: 16,120	48.8% and 36.9% in 90-day and 1-year mortality prediction improvements using ML compared with usual logistic regression data analysis
Kilic [62]	2021	ML approaches to analyzing adverse events following durable LVAD implantation	ENDURANCE trials Patients: 564	Bleeding, infection, and RHF are the most common postoperative adverse events. RHF has a strong transitive relationship with bleeding and infection
Nayak [63]	2022	ML algorithms that identify distinct phenotypes of RHF after LVAD implantation	IMACS data Patients: 2550	Four post-LVAD RHF phenotypes are identified Clinical outcomes are evaluated
Bahl [64]	2023	Explainable ML analysis of RHF after LVAD implantation	INTERMACS data Patients: 19,595	Five best predictors are identified Non-linear relationships are identified
Just [65]	2023	AI-based analysis of body composition that predicts the outcome for patients receiving long-term MCS	Preoperative CT scan Patients: 137	Adipose tissue is an indicator of postoperative major complications.

Abbreviations: RVF: right ventricular failure; LVAD: left ventricular assist device; INTERMACS: Interagency Registry for Mechanically Assisted Circulatory Support; PAP: pulmonary artery pressure; LDH: lactate dehydrogenase; RV EF: right ventricular ejection fraction; AUC: area under the ROC curve; RHF: right heart failure; ML: machine learning; CVP: central venous pressure; ROC: receiver operating characteristic; IMACS: International Registry for Mechanically Assisted Circulatory Support; MCS; mechanical circulatory support; CT: computed tomography.

## Data Availability

The data are provided within the manuscript.

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
