# Peer review of "The Role of Artificial Intelligence and Machine Learning in the Prediction of Right Heart Failure after Left Ventricular Assist Device Implantation: A Comprehensive Review"

_diagnostics, 2024, doi:10.3390/diagnostics14040380_

Round 1

Reviewer 1 Report

Comments and Suggestions for Authors

The paper is interesting but some changes are required.

The Abstract is too generic and not informative on the subject. Some details on the results and limitations of AI applications should be reported and discussed.

Introduction is too long and generic. 

In Table 1 the number of cases should be reported for each study. It should be also indicated if only a training set of retrospective patients was utilized for the studies or if the AI methods were also tested in a prospective evaluation set of new patients, to avoid overfitting. Which were the results of these prospective evaluation tests?

A similar Table reassuming the studies using conventional risk prediction scores should be also prepared for comparison.

Making the two Tables more informative could help in fact to compare the results of the different studies and reduce the length and complexity of paragraphs 2.1 and 2.2, which are very difficult to read and follow in the text.  

For AI studies in paragraph 2.2 It should be also indicated if only a training set of retrospective patients was utilized for the studies or if the AI methods were also tested in a prospective evaluation set of new patients, to avoid overfitting. Which were the results of these prospective evaluation tests?

At page 10 it is said : "While the researchers checked the area under curve  (AUC) they found that AUC of CRITT and PENN scores are 0,616 and 0,605 respectively.  The AUC of their AI systems is 0,729 which means the newly developed deep learning system is presented more accuracy to predict of RHF".

English of the phrase should be improved, but the main concern is that the improvement offered by the AI model seems really very low to the reviewer, and not significant and very disappointing from a clinical point of view. More than 25% of patients appear in fact not correctly classified even by the AI method. If this results is also achieved only in the training retrospective set of patients and not tested in a prospective manner, the risk of overfitting is very high and clinical usefulness appears very low. 

The same at page 13: .." The majority of AI studies achieved AUC rates  above 0.6 ".... 

These appear very disappointing results to me from a clinical point of view, since this means that around 40% of cases were not correctly classified by the AI methods. 

All these problems should be deeply discussed and comparison should be made among studies in the discussion and limitation sections.  

The limitations reported at paragraph 4, page 13, can not be accepted. The search of studies shoud be completed and extended to all the pertinent scientific fields, including paramedical publications, biomedical engineering and bioinformatics, related to the present topic. 

General limitations of AI, if considered as a black box, the certification of AI products as medical devices, legal implications of AI advice, the possibility of generalizable results in the heterogeneity of the clinical grounds and the real applicability of AI in the near future in this field shoud be also emphasized and discussed in more detail, taking into account pros and cons of the AI approach in the near future also in the considered setting of research.  

Comments on the Quality of English Language

Moderate editing is required.

Author Response

Comment/suggestion 0: The paper is interesting but some changes are required.

Our answer 0: We would like to thank the reviewer for their detailed and thought out evaluation as we took it seriously and agreed on many points, making required adjustments.

Comment/suggestion 1: The Abstract is too generic and not informative on the subject. Some details on the results and limitations of AI applications should be reported and discussed.

Our answer 1: More details of study results have been added to abstract. New and more details from discussion and conclusion were added to make abstract more informative.

Comment/suggestion 2: Introduction is too long and generic. 

Our answer 2: Introduction section is significantly shortened the and some changes made in structure.

Comment/suggestion 3: In Table 1 the number of cases should be reported for each study. It should be also indicated if only a training set of retrospective patients was utilized for the studies or if the AI methods were also tested in a prospective evaluation set of new patients, to avoid overfitting. Which were the results of these prospective evaluation tests?

Our answer 3: Number of cases added to table. There were no prospective studies. This fact was highlighted within discussion and conclusion sections and some comments were made about it. Both training and testing data were from retrospective patients.

Comment/suggestion 4: A similar Table reassuming the studies using conventional risk prediction scores should be also prepared for comparison.

Our answer 4: A new table about conventional risk prediction scores was added

Comment/suggestion 5: Making the two Tables more informative could help in fact to compare the results of the different studies and reduce the length and complexity of paragraphs 2.1 and 2.2, which are very difficult to read and follow in the text.  

Our answer 5: As you requested, placing them in a separate table to compare the conventional risk scores and AI studies helped us make comparisons easier and shorten paragraphs 2.1 and 2.2. Thank you for this guidance.

Comment/suggestion 6: For AI studies in paragraph 2.2 It should be also indicated if only a training set of retrospective patients was utilized for the studies or if the AI methods were also tested in a prospective evaluation set of new patients, to avoid overfitting. Which were the results of these prospective evaluation tests?

Our answer 6: Similar answer to Comment 3, this issue was discussed more within discussion and conclusion section of the review.

Comment/suggestion 7: At page 10 it is said : "While the researchers checked the area under curve  (AUC) they found that AUC of CRITT and PENN scores are 0,616 and 0,605 respectively.  The AUC of their AI systems is 0,729 which means the newly developed deep learning system is presented more accuracy to predict of RHF".

English of the phrase should be improved, but the main concern is that the improvement offered by the AI model seems really very low to the reviewer, and not significant and very disappointing from a clinical point of view. More than 25% of patients appear in fact not correctly classified even by the AI method. If this results is also achieved only in the training retrospective set of patients and not tested in a prospective manner, the risk of overfitting is very high and clinical usefulness appears very low. 

Our answer 7: The English of the specified section has been improved, as has the entire text. We added sections for discussion and results and commented on the AUC performance issue mentioned above.

Comment/suggestion 8: The same at page 13: .." The majority of AI studies achieved AUC rates  above 0.6 ".... 

These appear very disappointing results to me from a clinical point of view, since this means that around 40% of cases were not correctly classified by the AI methods. 

Our answer 8: Similar to comment 7, we made sure to highlight this performance issue that was brought to our attention by comments.

Comment/suggestion 10: All these problems should be deeply discussed and comparison should be made among studies in the discussion and limitation sections.  

Our answer 9: Similar to Comment 7, discussion of this issue was added.

Comment/suggestion 10: The limitations reported at paragraph 4, page 13, can not be accepted. The search of studies shoud be completed and extended to all the pertinent scientific fields, including paramedical publications, biomedical engineering and bioinformatics, related to the present topic. 

Our answer 10: Since the main subject of this review is right ventricular failure after left ventricular assist device applications, we particularly focused on the medical literature. However, the increasing role of artificial intelligence applications in the field of medicine suggests that we should also look at other scientific fields in the future.

Comment/suggestion 11: General limitations of AI, if considered as a black box, the certification of AI products as medical devices, legal implications of AI advice, the possibility of generalizable results in the heterogeneity of the clinical grounds and the real applicability of AI in the near future in this field shoud be also emphasized and discussed in more detail, taking into account pros and cons of the AI approach in the near future also in the considered setting of research.  

Our answer 11: As per this suggestion, a section about AI and its implications, current limitations and related discussions were added to conclusion

Reviewer 2 Report

Comments and Suggestions for Authors

This is an exhaustive review on predictive models for right ventricular failure following LVAD implantation. The review is systematic, with each article summarized to highlight its findings. In this regard, it is a useful article for those wishing to review the literature. However, I observe a series of structural problems that need to be addressed if the article is to be engaging for any reader interested in the topic.

Major issues:

  • Summarizing paragraph by paragraph and article by article without cohesion or highlighting characteristics makes the article monotonous and challenging to read. This repetitive structure is present in both the review of "classic" articles and those utilizing artificial intelligence. To address this, I recommend grouping articles, finding connections between them, providing tables, or incorporating figures. This would make the text more engaging and "readable."
  • The discussion is significantly shorter than the main text, and scientific advancement typically occurs in the discussion. I recommend restructuring it based on the mentioned connections and concluding by proposing parameters that the authors consider most important after this exhaustive review. Additionally, the initial paragraphs of the discussion literally "repeat" the introduction. Therefore, a complete overhaul of the discussion is necessary.
  • Perhaps this point repeats the previous ones, but a text that summarizes article after article and does not offer visual elements or aid in summarizing ideas is challenging to follow and unappealing. Besides understanding the topic, it is essential to present it in a way that captivates the reader's interest in learning more.

Minor issues:

  • Punctuation issues in the last paragraph of page 6.
  • Abbreviations in the first paragraph of page 7 (and the last paragraph of page 6).
  • The first sentence in the third paragraph of page 7 needs to be rewritten as it is not clear.
  • The mathematical formula in that paragraph should be redone or inserted as a figure.
  • In that paragraph, change "score of 12.5" to "score of 12.5 or higher" in the penultimate line.
  • The conclusion is very weak and does not contribute to the LVAD topic

Author Response

REVIEWER 2’s Comments/suggestions:

Comment/suggestion 0: This is an exhaustive review on predictive models for right ventricular failure following LVAD implantation. The review is systematic, with each article summarized to highlight its findings. In this regard, it is a useful article for those wishing to review the literature. However, I observe a series of structural problems that need to be addressed if the article is to be engaging for any reader interested in the topic.

Our answer 0: After recieving reviewer comments and suggestions, we took a critical look at our own work and agreed that structural issues within the text required extra attention and some sections needed extra content. Therefore we would like to express our gratitude for this critical review that was for the benefit of our final work.

Major issues:

  • Comment/suggestion 1: Summarizing paragraph by paragraph and article by article without cohesion or highlighting characteristics makes the article monotonous and challenging to read. This repetitive structure is present in both the review of "classic" articles and those utilizing artificial intelligence. To address this, I recommend grouping articles, finding connections between them, providing tables, or incorporating figures. This would make the text more engaging and "readable."
  • Our answer 1: We completely agree with your criticism on this matter. In order not to make the article monotonous with repetitive similar paragraphs, we shortened the sections for studies containing conventional risk scores and summarized the studies in a table. Thus, we made it both easier to understand and easier to compare.

  • Comment/suggestion 2: The discussion is significantly shorter than the main text, and scientific advancement typically occurs in the discussion. I recommend restructuring it based on the mentioned connections and concluding by proposing parameters that the authors consider most important after this exhaustive review. Additionally, the initial paragraphs of the discussion literally "repeat" the introduction. Therefore, a complete overhaul of the discussion is necessary.
  • Our answer 2: Discussion section restructured and added a significant amount of content based on suggestions.

  • Comment/suggestion 3: Perhaps this point repeats the previous ones, but a text that summarizes article after article and does not offer visual elements or aid in summarizing ideas is challenging to follow and unappealing. Besides understanding the topic, it is essential to present it in a way that captivates the reader's interest in learning more.
  • Our answer 3: In line with your suggestions, we tried to put the article in an easier to read and understandable format.

Minor issues:

  • Comment/suggestion 4: Punctuation issues in the last paragraph of page 6.
  • Our answer 4: Punctuation corrected
  • Comment/suggestion 5: Abbreviations in the first paragraph of page 7 (and the last paragraph of page 6).
  • Our answer 5: Abbreviations full form added within text
  • Comment/suggestion 6: The first sentence in the third paragraph of page 7 needs to be rewritten as it is not clear.
  • Our answer 6: Sentence was rewritten as suggested
  • Comment/suggestion 7: The mathematical formula in that paragraph should be redone or inserted as a figure.
  • Our answer 7: Mathematical formula summarized and restructured within the newly added table
  • Comment/suggestion 8: In that paragraph, change "score of 12.5" to "score of 12.5 or higher" in the penultimate line.
  • Our answer 8: "score of 12.5" is changed to "score of 12.5 or higher" in the penultimate line.
  • Comment/suggestion 9: The conclusion is very weak and does not contribute to the LVAD topic

Our answer 9: Conclusion section is significantly changed and sizeable new content added.

Reviewer 3 Report

Comments and Suggestions for Authors

The authors do not clarify whether their review has been systematic or not. In any case, it is worthwhile to expand the review and try to be as systematic as possible. I further recommend the following:

The introduction should be shortened.

In Figure 1, the font should be enlarged as it is barely readable.

Make a table with the reviewed papers to evaluate the conventional risk prediction scores.

The results should be shortened and the discussion should be expanded (some results information really belongs to discussion).

Limitations should include that the review is not systematic.

Author Response

REVIEWER 3’s Comments/suggestions:

Comment/suggestion 0: The authors do not clarify whether their review has been systematic or not. In any case, it is worthwhile to expand the review and try to be as systematic as possible. I further recommend the following:

Our answer 0:  After reading and understanding the feedback provided with your suggestions, we realized that some sections required improvements and attempted to rectify them. Suggestions and comments were most welcome.

Comment/suggestion 1: The introduction should be shortened.

Our answer 1: The introduction has been significantly shortened.

Comment/suggestion 2: In Figure 1, the font should be enlarged as it is barely readable.

Our answer 2: Figure was corrected with larger fonts as suggested

Comment/suggestion 3: Make a table with the reviewed papers to evaluate the conventional risk prediction scores.

Our answer 3: Conventional risk prediction scores and related papers added in a new table

Comment/suggestion 4: The results should be shortened and the discussion should be expanded (some results information really belongs to discussion).

Our answer 4: Discussion section is significantly expanded and  the changes are made as suggested.

Comment/suggestion 5: Limitations should include that the review is not systematic.

Our answer 5: A statement about the review not being systematic is added to limitations.

Round 2

Reviewer 1 Report

Comments and Suggestions for Authors

The Authors made the required changes.

Comments on the Quality of English Language

Minor editing required.

Reviewer 2 Report

Comments and Suggestions for Authors

It is still a bad article, in the sense that it contributes nothing to the topic. They have adequately responded to all open queries, and, although the quality of the article has increased, I still do not see it as attractive to the general public. Due to the effort made, I recommend accepting, but it is the editor who must decide if such a specific and unnew article should be published in JCM